

# How to reduce sampling errors in spaceborne cloud radar-based snowfall estimates

Filippo Emilio Scarsi[1,2], Alessandro Battaglia[1,3], Maximilian Maahn[4], and Stef Lhermitte[5,6]

[1]Politecnico of Torino, Torino, Italy
[2]University School for Advanced Studies IUSS Pavia, Pavia, Italy
[3]University of Leicester, Leicester, UK
[4]Leipzig University, Leipzig, Germany
[5]KU Leuven, Leuven, Belgium
[6]TUDelft, Delft, Netherlands

**Correspondence:** Filippo Emilio Scarsi
filippo.scarsi@polito.it

**Abstract.** Snowfall is an important climate change indicator affecting surface albedo, glaciers, sea ice, freshwater storage, cloud lifetime and ecosystems. Precise snowfall measurements at high latitudes are particularly important for the estimation of the mass balance of ice sheets; however, the snowfall is difficult to quantify with in-situ measurements in those locations. In this context, spaceborne radar and radiometers atmospheric missions can help in the assessment of snowfall at high latitudes.

The decommissioned NASA CloudSat mission provided invaluable information about global snowfall climatology from 2006 to 2023. The CloudSat-based estimates of global snowfall are considered the reference for global snowfall estimates, but these data suffer from poor sampling and the inability to see shallow precipitation, which limits their use, for example, as input to surface mass balance models of the major ice sheets. WIVERN (WInd VElocity Radar Nephoscope), one of the ESA Earth Explorer 11 candidate missions (final selection in July 2025), is equipped with a conical scanning 94 GHz Doppler radar

and a passive 94 GHz radiometer, with the main objective of measuring global in-cloud horizontal winds, but also quantifying cloud ice water content and precipitation rate. Its conically scanning system, with a 42° incidence angle is expected to reduce the radar blind zone near the surface (especially over the ocean) and allows the mission to have a swath width of 800 km and 70 times more sampled points than a fixed looking instrument. This radar measurements tackle the current uncertainties in snowfall estimates, highly improving the sampling frequency and accuracy of snowfall measurements.

The uncertainty in snowfall measurements arises from various factors, including the diurnal cycle, uncertainty in the Z-S relationship and the sampling error. This study quantifies each of these contributors individually and demonstrates the improved sampling capabilities of the WIVERN conically scanning geometry for some specific regions (Antarctica, Greenland) by computing the sampling error at different spatial and temporal scales via simulations of WIVERN vs. CloudSat orbits and scanning geometry, based on the snowfall rates produced by ERA5 reanalysis.

Results show that a WIVERN-like conically scanning system significantly reduces the uncertainty in polar snowfall estimates, if compared to a CloudSat-like near nadir fixed viewing geometry. While CloudSat generates acceptable errors at the annual zonal scales, WIVERN can produce estimates within the climatological variability for latitude-longitude domain larger





than $0.5° \times 0.5°$ already at the 10-day timescale, making it a valuable product for regional climate model evaluation and as an input to surface mass balance models of the major ice sheets and glaciers.


# 1 Introduction

In polar regions and mid-latitudes, most precipitation is formed through the ice phase as snowfall (Mülmenstädt et al., 2015). For high latitudes and mountainous regions, it is the dominating form of precipitation at the ground (Field and Heymsfield, 2015). Therefore, snowfall removes not only moisture form the atmosphere but plays a crucial, interlinking role in the climate
system. In the cryosphere, snowfall is the only mass source term for glaciers and ice sheets, and thus crucial for their mass balance (Souverijns et al., 2018a). On sea ice, snow forms an insulating layer between sea ice and atmosphere impacting sea ice lifetime (Perovich et al., 2017). On land, snow modifies the surface albedo relevant for the ice albedo feedback (Hall, 2004). Furthermore, snow cover impacts ecology (Slatyer et al., 2022), traffic safety (Strong et al., 2010), recreation (Steiger et al., 2019), and freshwater storage which is also relevant for hydropower generation (Wasti et al., 2022). In a warming
climate, precipitation amounts and extreme events, including heavy snowfall, are expected to increase (Quante et al., 2021), but the estimates of the exact magnitudes are affected by large uncertainties (Lopez-Cantu et al., 2020). This is because the exact pathways through which ice particles, liquid water, cloud dynamics, and aerosol particles are interacting during snow formation are not well understood (Morrison et al., 2012; Griesche et al., 2021).

Better observations of the fingerprints of snowfall formation processes at sufficient spatio-temporal resolution are needed
to advance our understanding of ice and mixed-phase clouds and precipitation formation processes (Morrison et al., 2020). Traditionally, snowfall is measured with in situ gauges, but high spatial variability (Scipión et al., 2013), poor coverage (Kidd et al., 2017), and wind-related under-catch (Yang et al., 1999) pose significant challenges.

The deficits of in situ snowfall observations requires using remote sensing techniques. Because ground-based remote sensing with weather radar is available only in densely populated areas and few sites are equipped with radars in the polar regions (e.g.,
Matrosov et al. (2008); Souverijns et al. (2018b); Li et al. (2021); Schoger et al. (2021); Matrosov et al. (2022); Tridon et al. (2022); Alexander et al. (2023)) space-borne remote sensing techniques are the prime method to observe snowfall globally. Passive microwave sensors such as AMSU (Advanced Microwave Sounding Unit) offer good spatial coverage due to their km-scale imaging capabilities, but passive microwave signals are also impacted by surface properties (Chen and Staelin, 2003; Skofronick-Jackson et al., 2004; Skofronick-Jackson and Johnson, 2011) and the presence of supercooled liquid water (Wang
et al., 2013; Battaglia and Panegrossi, 2020; Panegrossi et al., 2022) which are difficult to separate from atmospheric scattering contributions by frozen hydrometeors.

Due to their unique profiling capabilities, radar can provide profile properties of hydrometeors and separate scattering by hydrometeors from the surface. Even though the conversion into snowfall rates is associated with uncertainties related to the



indirect observation, space-borne radars on low orbit satellites such as CloudSat (Stephens et al., 2002) and EarthCare (Wehr
et al., 2023) provide the best way to observe snowfall globally (Milani and Kidd, 2023). CloudSat snowfall measurements
have been successfully evaluated with ground based in situ (Hiley et al., 2011) and ground based radar networks (Smalley
et al., 2014; Mroz et al., 2021). The data has been used to produce snowfall climatologies (Liu, 2008; Bennartz et al., 2019;
Kulie et al., 2020) which are most relevant in regions with sparse in situ observations such as Antarctica or Greenland. Further,
CloudSat data was used to study seasonal cycles (Kulie and Milani, 2018), evaluate climate models (Palerme et al., 2017),
and to study the surface mass balance of ice sheets (Boening et al., 2012; Milani et al., 2018). However, the combination of
CloudSat's revisit time of 16 days combined with the 1 km footprint of the observations leads to a sparse spatial sampling,
causing noise in snowfall climatologies even when averaging over 10 years (Kulie et al., 2020). Further, it was found that
CloudSat's snowfall retrieval has biases for snowfall rates exceeding $1.0 \, \mathrm{mm \, h^{-1}}$ (Cao et al., 2014; Norin, 2015). Due to
surface clutter contamination, CloudSat cannot observe snowfall in the blind zone which is up to 1200 m thick and can lead to
an underestimation of snowfall rate for shallow events but also to an overestimation of snowfall rate due to sublimation (Maahn
et al., 2014).

In this study, we will show the potential of the ESA Earth Explorer 11 candidate mission WIVERN (WInd VElocity Radar
Nephoscope, www.wivern.polito.it, Illingworth et al. (2018); Battaglia et al. (2018, 2022); ESA (2023); Rizik et al. (2023);
Tridon et al. (2023)) global snowfall monitoring. Different to CloudSat and EarthCare, WIVERN's cloud radar will not measure
at nadir but will scan conically at 38° angle (for measuring horizontal in-cloud wind) and also feature a 94 GHz passive channel.
While WIVERN is also expected to feature a smaller blind zone over ice-free ocean and has the potential for improved snowfall
retrievals due to the availability of a passive channel (Battaglia and Panegrossi, 2020), we will focus on how the conical
scanning, with 70 times better coverage than for a nadir pointing instrument such as CloudSat or EarthCare, improves snowfall
estimates. WIVERN and the methodology are introduced in Section 2, results are presented in Section 3, and concluding
remarks are given in Section 4.

## 2 Methodology

The basis for this work is the ERA 5 hourly surface snowfall reanalysis product (Hersbach et al. (2023), last access: 15 March
2024), with a spatial resolution of $0.25° \times 0.25°$ for a total time span of 20 years from 2001 to 2020. We use it as a reference for
comparison with the accumulated snowfall as retrieved by a WIVERN-like and a CLOUDSAT-like radar instrument based on
the same ERA 5 dataset. The sampling of the radar footprints have been computed based on the viewing geometry (see Tab. 1)
and the satellite orbits, which have been propagated in the period of interest according to the orbital parameters reported in
Tab. 1. Then, for each time stamp of the selected ERA5 dataset, a mask that indicates whether any given geolocated snowfall
dataset $0.25° \times 0.25°$ grid point is sampled by the instrument is generated according to the radar footprints positions at ground.
With a conically scanning radar, several passes over the same grid point may occur within minutes, but we count several passes
within an hour as one.



**Table 1.** WIVERN and CloudSat (Stephens et al., 2002) orbit and radar specifics. The shown configuration of WIVERN is the one currently under Phase-A study for the ESA Earth Explorer 11 program.

| Satellite | WIVERN | CloudSat |
|---|---|---|
| Spacecraft height, $H_{SC}$ | 500 km | 705 km |
| Spacecraft velocity, $v_{SC}$ | 7600 ms$^{-1}$ | 7600 ms$^{-1}$ |
| Orbit inclination, $i$ | 97.42° | 98.2° |
| Orbit Local Time of the ascending node, $LTAN$ | 06:00 | 13:30 |
| Orbit repeat cycle | 5 days | 16 days |
| Off-nadir pointing angle | 38° | 0.16° |
| Swath width at ground | 800 km | 1.4 km |
| Radar output frequency | 94.05 GHz | 94.00 GHz |
| Radiometer channel | 94 GHz | - |
| Antenna angular velocity, $\Omega_a$ | 12 rpm | – |
| Footprint speed | 500 kms$^{-1}$ | 7 kms$^{-1}$ |
| Minimum detectable reflectivity | -21 dBZ | -28 dBZ |

The mask has been applied to the ERA5 snowfall dataset to produce two datasets, with the snowfall simulated as observed by the CloudSat and WIVERN instruments using the following procedure. The hourly ERA5 snowfall, $S$, is converted to the equivalent radar reflectivity factor $z_e$ according to a mean climatological relationship as proposed by Hiley et al. (2011) through

$$z_e = a_{mean} S^{b_{mean}} \qquad a_{mean} = 21.6, \quad b_{mean} = 1.2 \qquad (1)$$


with $z_e$ in mm$^6$ m$-3$. Typically, reflectivity is used in logarithmic units dBZ converted with $Z_e = 10 \times \log_{10} z_e$. The reflectivity below the nominal radar sensitivity ($-21$ dBZ and $-28$ dBZ for WIVERN and CloudSat, respectively) is set to 0 mm$^6$ m$-3$, as below these thresholds, no snow precipitation is expected to be detected. The $Z_e$ values are converted to snowfall rate by introducing random noise associated with the uncertainty in the $Z_e - S$ relationship. For this, $S$ is sampled from a log-normal distribution whose mean value corresponds to the ERA5 value and whose standard deviations are computed as half the


difference $S_{1\sigma} - S_{-1\sigma}$ where $S_{1\sigma}$ and $S_{-1\sigma}$ are assumed equal to:

1. $S_{+1\sigma} = 0.0238 \, z_e^{0.909}$;

2. $S_{-1\sigma} = 0.21 \, z_e^{0.769}$.

which are the inverse formulas of $z_e = 61.2 \, S^{1.1}$ and $z_e = 7.6 \, S^{1.3}$, respectively, proposed by Hiley et al. (2011) as lower and


upper boundaries for the uncertainty in the $Z_e - S$ relationship. This represents a worst-case estimate of the uncertainty caused by the $Z_e - S$ relationship, as we assume it varies randomly from grid box to grid box, whereas in reality it may be spatially



correlated. In this study, we neglect errors related to the fact that $S$ is not observed at the surface, but at an higher altitude due to the surface clutter (1200 m for CloudSat, Maahn et al., 2014).

Fig. 1 shows an example case of geolocated ERA5 snowfall rate (January 2, 2020 18:00 UTC) in comparison to the cor-
responding simulated WIVERN and CloudSat retrievals. Despite its sparse sampling within its 800 km swath, the WIVERN footprint samples all $0.25°$ grid points within the swath, with an obvious benefit compared with the CloudSat pencil beam.

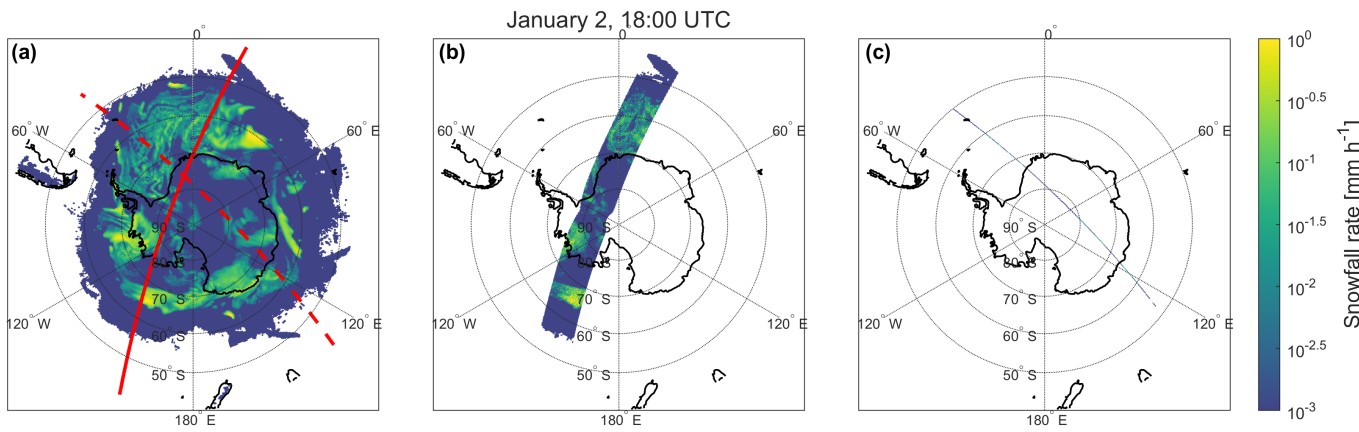

**Figure 1.** Example of a geolocated WIVERN and CloudSat snowfall rate retrieval obtained at a given hour. Panel (a) shows the geolocated ERA5 accumulated snowfall rate at January 2, 2020 at 18:00 UTC, with the satellites' groundtrack of WIVERN and CloudSat groundtrack outlined with the solid and dashed red lines, respectively. Panels (b) and (c) depict what would be the corresponding snowfall rate retrieval of WIVERN and CloudSat, respectively. Uncertainty due to application of the $Z_e - S$ relationship has been included.

Finally, the snowfall retrieved in each $0.25° \times 0.25°$ grid-box is aggregated at different time scales (e.g. a month, a year). The results can then be further aggregated over coarser spatial domains.

The simulated snowfall retrievals from the satellites are compared with the ERA5 reference dataset to assess the reliability of
the WIVERN ($S_{WIV}$) and CloudSat ($S_{CS}$) data for estimating snowfall accumulation at different spatial and temporal scales.

For each investigated spatial and temporal resolution, time series data is accumulated for a total of 20 years. From the three time series of $S_{ERA5}$, $S_{WIV}$ and $S_{CS}$, the bias ($AB_{WIV/CS}$) the root mean square error ($RMSE_{WIV/CS}$) and their



normalised counterparts ($NAB_{WIV/CS}$, $NRMSE_{WIV/CS}$) are estimated with

$$AB_{WIV/CS} = \frac{1}{N}\sum_{i=1}^{N}\left|S_{WIV/CS}[i] - S_{ERA5}[i]\right| \tag{2}$$

$$NAB_{WIV/CS} = \sum_{i=1}^{N}\left(\left|S_{WIV/CS}[i] - S_{ERA5}[i]\right| / \sum_{i=1}^{N}S_{ERA5}[i]\right) \tag{3}$$

$$RMSE_{WIV/CS} = \sqrt{\frac{1}{N}\sum_{i=1}^{N}\left(S_{WIV/CS}[i] - S_{ERA5}[i]\right)^2} \tag{4}$$

$$NRMSE_{WIV/CS} = \sqrt{\frac{\sum_{i=1}^{N}\left(S_{WIV/CS}[i] - S_{ERA5}[i]\right)^2}{\sum_{i=1}^{N}\left(S_{ERA5}[i]\right)^2}}. \tag{5}$$

The differences between the simulated fields and the ERA5 reference is mainly driven by three factors: the radar sensitivity leading to the omission of low-reflectivity events, the uncertainties in the $Z_e - S$ relationship and the instrument sampling (i.e., the fact that at any location $S$ is sampled intermittently according to the revisit time). The latter contribution can be further decomposed into the contribution associated with the diurnal cycle of the orbit (i.e. the fact that at any given location the satellite passes only at certain times of the day) and to the sparseness of the measurements on different days.

## 3 Results

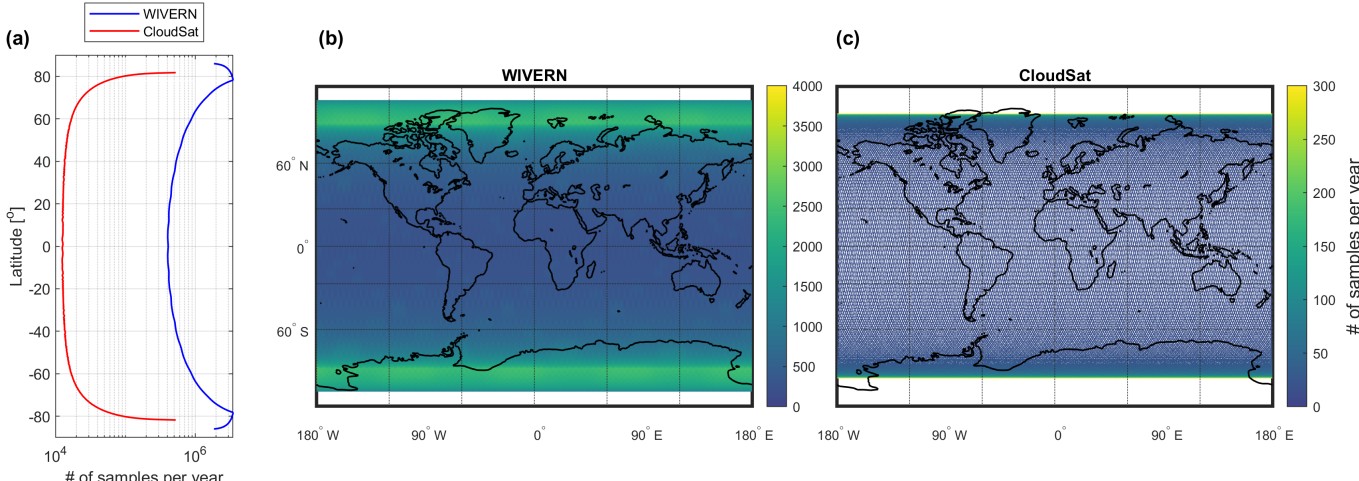

**Figure 2.** Number of annual samples collected by WIVERN (b) and CloudSat (c) per $0.25° \times 0.25°$ grid box. Note different color scales for the two panels. Zonal overpasses averaged over $0.25°$ wide latitude bands are shown in panel (a).





The number of annual samples collected by WIVERN and CloudSat shown in Fig. 2 clearly demonstrates the advantage of
the WIVERN sampling with an average of $6.696 \times 10^8$ total number of samples per year compared to $1.45 \times 10^7$ for CloudSat.
While WIVERN produces global coverage with a resolution of $0.25°$ resolution for absolute latitudes below $86°$, CloudSat has
gaps (white spots in the right panel of Fig. 2) due to its periodic orbit and its viewing geometry. The WIVERN reduction of
the blind region near the Poles up to $86°$ latitudes means a coverage of $95\%$ of the Antarctic continent, which is a significant
improvement in comparison to CloudSat's coverage of only $75\%$ of the continent.

**Figure 3.** Panel (a) shows the mean annual accumulated snowfall according to ERA5 from 2001 to 2020. Panel (b) shows the normalized
standard deviation, hence the inter-annual variability of the snowfall. The corresponding normalized root mean squared error on the 1-year
accumulated snowfall sampled by WIVERN and CloudSat is also shown in panels (c) and (d), respectively.

We set the estimated CloudSat and WIVERN errors in perspective to the mean snowfall rate. Fig. 3 shows the mean annual
accumulated snowfall according to the ERA5 dataset in panel (a) and the normalized inter-annual variability of such snowfall





in panel (b). The figure also depicts the NRMSE of the WIVERN (panel (c)) and CloudSat (panel (d)) annual accumulations. For WIVERN, the NRMSE is lower than 0.5 for most regions, with higher values in regions where the snowfall is rare so that the number of samples is low (i.e. in lower latitudes). The NRMSE tends to decrease when moving toward the poles due to the improved sampling (Fig. 2) and less intermittent snowfall observations. For CloudSat, the NRMSE is almost everywhere above 0.5, rising to much higher values in regions where the snowfall is very rare, due to the strong intermittency of the phenomenon and the poor sampling. The NRMSE constantly decreases as the satellite approaches the polar regions due to the higher number of samples collected by the satellite and the high sensitivity of the CloudSat CPR. Furthermore, the nadir-looking viewing geometry of CloudSat CPR together with the repetition of the satellite's ground-track, generates gaps in the sampling of increased size as they get closer to the equator (see panel (d) in Fig. 3 or panel (c) in Fig. 2).

## 3.1   Errors on snowfall accumulation at different spatial and temporal scales

In order to answer the question of how the error varies when the temporal scale of accumulation is changed, the analysis has been conducted examining the estimated annual, monthly and 10-day accumulated snowfall. Similarly, for spatial scales, global snowfall has been aggregated into grids with lat×lon box sizes of $0.25° \times 0.25°$, $0.5° \times 0.5°$, $1° \times 1°$, $5.0° \times 5.0°$ and $10.0° \times 10.0°$. Zonal averages with a latitude resolution of $0.5°$ have been studied to observe the zonal mean behavior of the error as well.

Fig. 4 depicts the WIVERN (blue lines) and CloudSat (red lines) NRMSE for different temporal and spatial scales binned by snowfall rate. As highlighted in Fig.4, the error, and hence the measurement uncertainty, is mitigated when averaging over higher spatial domains (i.e. left vs right figure columns) and longer temporal scales (i.e. upper vs lower figure rows). The error/uncertainty improvement is more pronounced in CloudSat than in WIVERN, as the latter has a much better sampling baseline. The separation between WIVERN and CloudSat also decreases with increasing snowfall rate. In fact, regions with low snowfall rates are typically characterized by rare snowfall events, which are likely to be missed by the poor sampling of CloudSat. At the annual and monthly scales, the CloudSat NRMSE for zonal snow is lower than the climatological variability (with exceptions at very low snowfall rates). However, when looking at CloudSat $5° \times 5°$, the error exceeds the variability and is comparable to the WIVERN error at a much finer scale ($0.25° \times 0.25°$).

If the climatological variability is used as a threshold for acceptable measurement uncertainty, then CloudSat annual accumulations can only be used at the annual zonal domains. Vice versa, WIVERN produces errors lower than the natural variability at domains of at least $0.5° \times 0.5°$. WIVERN $0.25° \times 0.25°$ can still be useful but only for annual and monthly accumulations larger than 864 mm and 108 mm, respectively.

It is important to note that at spatial scales larger than $0.25° \times 0.25°$, the domain in which we are trying to estimate the snowfall is very large if compared to the CloudSat swath width (1.4 km at ground level). Therefore, with CloudSat, snowfall estimation in such areas is done with a small number of points, which are not very representative of the total snowfall in the area. This effect is accentuated at the 10-day timescale and mitigated for monthly and annual accumulations. Furthermore, CloudSat's orbit repetition time is 16 days, which means that full coverage in not achieved in 10 days.





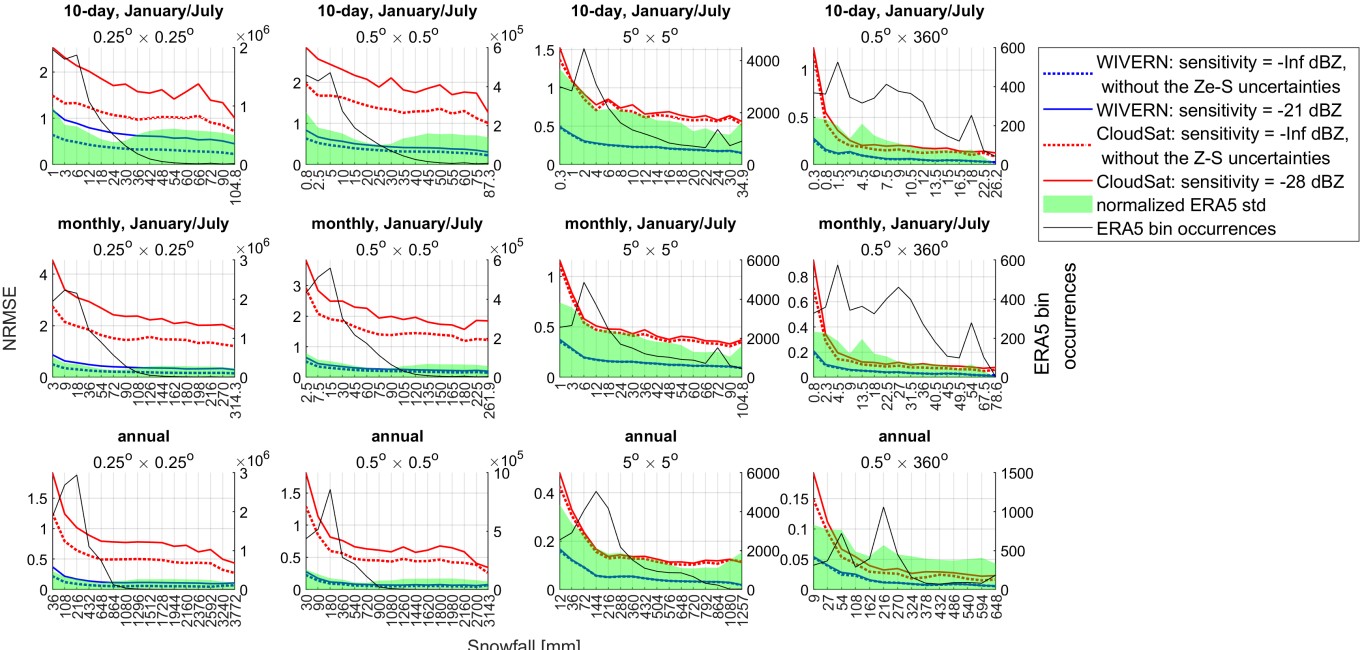

**Figure 4.** NRMSE for WIVERN (blue lines) and CloudSat (red lines) as a function of different snowfall classes and for different $lat \times lon$ grid box sizes and the zonal mean. The NRMSE considering only the sampling contribution (dashed lines) and all sources of error (solid lines) are shown with different line styles. The 10-day, monthly and annual time scales are shown in the top, middle and bottom row, respectively. The snowfall classes are defined as snowfall intervals; e.g., for the annual timescale, the first bin corresponds to the interval of snowfall between 36 and 108 mm and the last bin between 3772 mm and infinity. Results for the 10-day and monthly cases are shown when considering data of January in the Northern Hemisphere, and July in the Southern Hemisphere.

In order to provide useful benchmarks, WIVERN (or CloudSat) NRMSE must be lower than the normalized climatological variability of ERA5 snowfall (defined as the temporal standard deviation), which is indicated by the green shaded area. The solid black line indicates the number of occurrences in the ERA5 analyzed dataset for the specific class (with the $y-$axis scale drawn on the right side).

As highlighted by Roberts et al. (2018), it is important to have precipitation datasets with spatial resolution better than 100 km poleward of $50°$; the WIVERN mission could significantly contribute to such goal by providing snowfall rates at spatial scales better than $0.5°$.

### 3.1.1 Impact of sampling error

The sampling error associated with the intermittent sampling of the snowfall (Fig. 4) is typically the dominant source of error;
it decreases if the number of samples increases e.g. when coarsening the spatial and/or the temporal scale and/or if the snowfall becomes less intermittent (e.g., typically for higher accumulations). As the WIVERN sampling is much better than that of CloudSat, its sampling error is always lower than that of CloudSat (by at least a factor of two).





WIVERN and CloudSat orbits are both sun-synchronous, with a mean local time of the ascending node (MLTAN) of 06:00 AM and 01:45 PM, respectively. This means that for any given point on the Earth's surface, the spacecraft will always pass over

that point at the same local time. The local time of the observation is the same as the local time of the satellite overpass. This is also true for WIVERN; however, the large swath width means that the same point can be observed at different local times, especially at high latitudes (e.g. at $80°S$ latitude there are on average 6.4 samples per day, see Fig. 2). Sampling a given site at only a few specific times of the day introduces an error in the snowfall accumulation due to the snowfall diurnal cycle (Watters et al., 2021; Milani and Wood, 2021), which is part of the sampling error. Since for WIVERN measurements at latitudes above

$60°N$ and $60°S$, the maximum revisit time (worst case scenario) is always less than 1 day (Battaglia et al., 2022), WIVERN sampling errors are only induced by the diurnal cycle effect. However, this is not the case for CloudSat CPR sampling which is characterized by an orbit repetition time of 16 days.

WIVERN's sampling errors are always smaller than the climatological variability at any spatial and temporal scales. Conversely, averaging over domains larger than $5°×5°$ is required at all timescales to reduce CloudSat sampling errors below

the threshold dictated by the natural variability, with the sampling errors on the zonal snowfall being comparable with the WIVERN sampling errors for domains $0.25°×0.25°$ in size.

### 3.1.2 Impact of the radar sensitivity

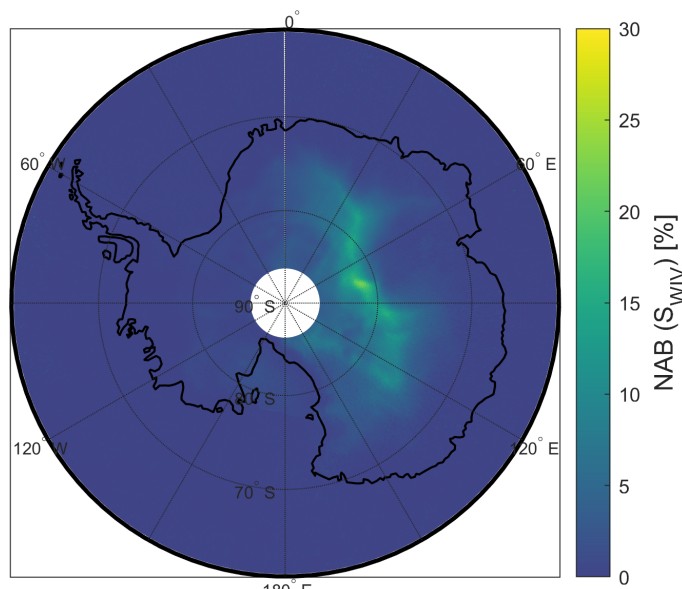

**Figure 5.** Normalized absolute bias between $S_{WIV}$ computed with sensitivity = -21 dBZ, and $S_{WIV}$ computed without the error induced by the sensitivity, normalized by the latter.





The effect of the sensitivity emerges at locations where the snowfall rates generate reflectivities below the sensitivity of the radar. When adopting the $Z_e - S$ relationship of Eq. (1) the minimum detectable snowfall rate is $7.9 \times 10^{-3}$ and $1.6 \times 10^{-3}$ mm for WIVERN and CloudSat, respectively. Due to WIVERN's worse sensitivity, this effect is more pronounced than for CloudSat, and is only really significant only for specific regions where snow rates below the detection threshold contribute significantly to the total accumulation. In particular, the error of the WIVERN accumulated snowfall in the region of the Antarctic desert comprised between $0°$ and $150°E$ is strongly affected by this source of error, as it can be seen in Fig. 3 and Fig. 5. Other regions such as central Greenland and western China are affected as well. However, globally, or when looking at the snowfall zonal behavior depicted in Fig. 6, or at points clustered based on similar snowfall values as outlined in Fig. 4, this effect appears to be negligible.

### 3.1.3 Impact of uncertainties in $Z_e - S$ relationship

Snowfall retrievals, especially those based on a single frequency, are limited by various uncertainties such as the characterization of the snowflake size distribution and the modeling of the backscattering properties of the ice crystals (Hiley et al., 2011; Kneifel et al., 2020; Tridon et al., 2019). At large snowfall rates, non-Rayleigh effects at the 94 GHz band cause further problems in the estimation of the snowfall rate. Uncertainties associated with the retrieval of $S$ from $Z_e$ are considered in this study as described in section 2, but it is important to note that the estimate of the $Z_e - S$ is assumed to be unbiased.

Fig. 4 shows the contributions of the sampling error, the $Z_e - S$ uncertainty and the sensitivity to the total error. As the latter contribution is negligible, the difference between the sampling error and the total error highlights the importance of the $Z_e - S$ uncertainties in the snowfall retrieval. For both WIVERN and CloudSat, the total NRMSE almost doubles compared to when considering only the effect of sampling at finer spatial scales, such as for the grid box size of $0.25° \times 0.25°$. Instead, when averaging the snowfall on larger areas, e.g. increasing the size of the grid boxes, the impact of the $Z_e - S$ is strongly mitigated, as expected from the assumption of the $Z_e - S$ estimate being unbiased. For WIVERN, thanks to the high number of collected samples, the contribution of the $Z_e - S$ uncertainty becomes negligible starting from $1.0° \times 1.0°$ spatial scale (not shown).

### 3.2 Errors on zonal snowfall: from annual to 10-day scales

For global precipitation studies, zonal precipitation estimates are of particular interest for the observation of the Earth's climate, the detection of climate change and to evaluate and constrain historical and future climate simulations (Hagemann et al., 2006; Hegerl et al., 2015; Egli et al., 2022).

The zonal mean snowfall, where the latitude resolution is $0.5°$, is shown in Fig. 6. WIVERN and CloudSat can capture the zonal climatological mean of the reference at the monthly and annual timescales (not shown), with the second being a bit more noisy than the first. CloudSat RMSE is within the standard deviation of ERA5 only for annual means. At the monthly scale, CloudSat exceeds the standard deviation of ERA5 in the Northern Hemisphere during the warm season between $60°N$-$65°N$ and $25°N$-$40°N$, and during the cold season between $25°N$-$60°N$ and $25°S$-$60°S$. For the inter-annual variability at annual timescale, CloudSat exceeds the natural variability only at latitudes between $25°N$-$45°N$ and $25°S$-$45°S$. Instead, WIVERN





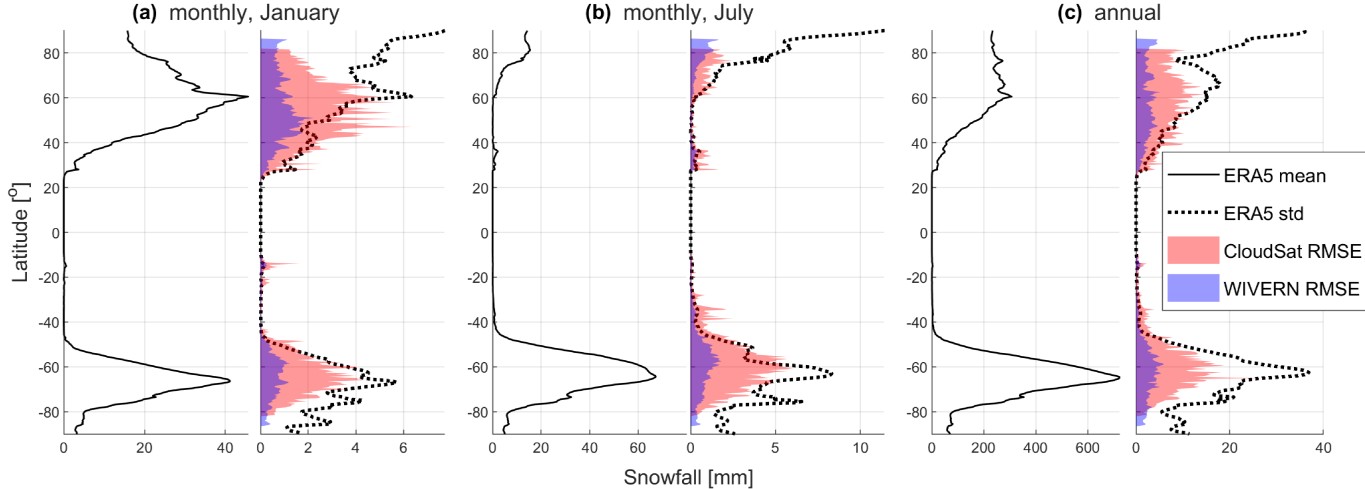

**Figure 6.** The figure shows the zonal mean snowfall at the monthly (January, panel (a) and July, panel (b)) and annual (panel (c)) timescales. The ERA5 mean value is depicted with a solid black line. WIVERN and CloudSat mean values are not shown, and they are indistinguishable from the ERA5 mean. The inter-annual variability of the zonal mean (i.e. ERA5 standard deviation) is shown by the black dashed line. The shaded areas outline the RMSE of WIVERN and CloudSat.

220  RMSE remains within the climatological variability not only at annual and monthly scales, but also at the 10-day scale (not shown).

## 3.3  Regional estimation of accumulated snowfall

The estimation of the snowfall in polar regions is of primary importance for quantifying the ice sheet mass balance and monitoring potential ice loss. Therefore, an analysis to quantify the regional effects of CloudSat and WIVERN sampling has

225  been carried out in regions of Antarctica and Greenland defined by their drainage systems Zwally et al. (2012). In Zwally et al. (2012), each basin is assigned an ID number and the subdivision is shown Fig. 7. Antarctic regions of particular interests are the following:

– the Amundsen Sea sector, which consists of basins ranging from 20 to 22: it is characterized by the strongest ice mass loss on the continent, as described in Yang et al. (2023).

230  – the Antarctic Peninsula, which consists of basins ranging from 24 to 27, has experienced a rapid warming in recent years.

Such regions are also characterized by a large snowfall accumulations.

For Greenland, when considering the loss of ice sheet mass, the conditions are less variable over the area (Mouginot et al., 2019) and the regions of interests correspond to the basins 3.3, 4.1, 4.2, 4.3 and 5.0, which are the ones affected by the highest snowfall precipitation (see Fig. 3).



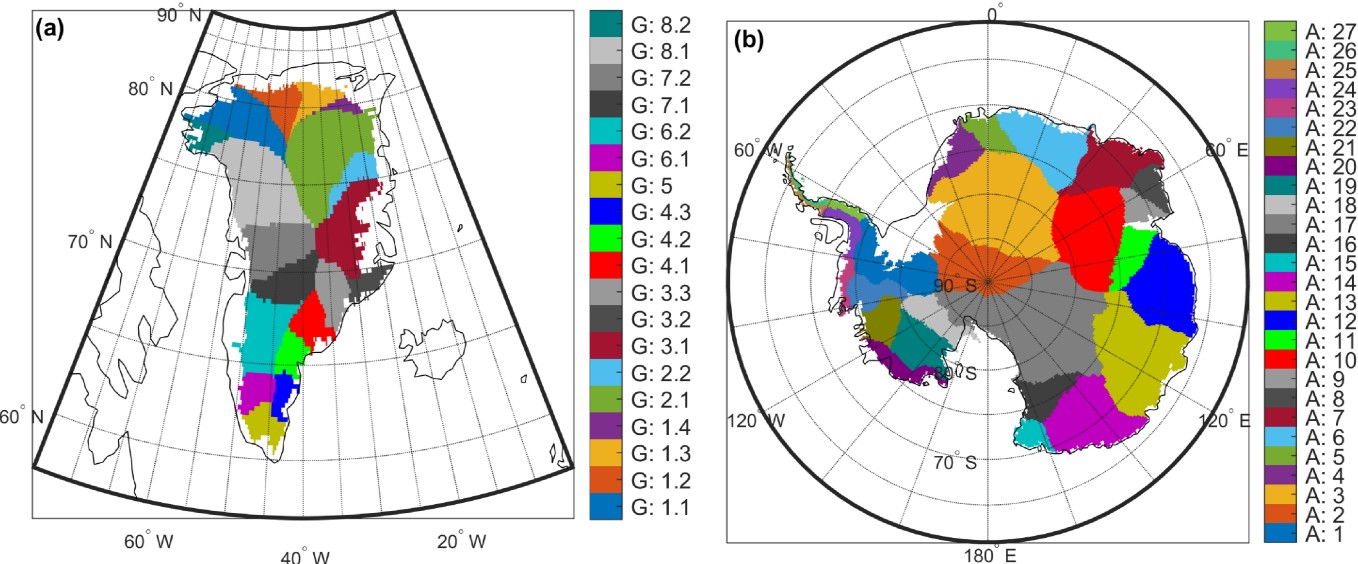

**Figure 7.** Division of Greenland and Antarctica into sub-regions based on the basins, according to Zwally et al. (2012).

In order to provide useful measurements for estimating the total snowfall accumulation in a certain region, the RMSE must be significantly lower than the climatological variability of the region. As shown in Fig. 8, when trying to estimate the total snowfall in the regions of Antarctica, both WIVERN and CloudSat measurements can provide very useful benchmarks at all time scales, as their RMSE is low compared to the climatological variability of the regions, with some exceptions for CloudSat (regions 1, 2, 17 and 26 at all timescales, 5 and 25 at the 10-day scale, and 27 at 10-day and monthly scales).

When examining regions of Greenland, as shown in Fig. 9, while WIVERN generally provides useful measurements for estimating total snowfall, CloudSat's poor sampling results in a very large RMSE, strongly exceeding ERA5 variability in a lot of cases: 3,3, 4.1, 4.2, 4.3, 5.0 and 6.1 at all the time scales, and 3.2 at the 10-day and monthly scales, and 6.2 at the 10-day scale.

Overall, WIVERN produces a significantly lower RMSE than CloudSat (lower by at least a factor of 2), indicating that WIVERN can provide more reliable estimates of regional snowfall variability.

Biases in the mean snowfall are introduced by the sampling, indicating an overestimation or underestimation of the snowfall, and are larger in Greenland than in Antarctica. Overall, WIVERN produces smaller biases than CloudSat, with some exceptions (e.g. region G:4.3 at the annual scale) related to the sensitivity. The number of samples collected by both satellites is higher in Antarctica than in Greenland (see Fig. 2), causing the RMSE and the bias to be larger in the latter.

WIVERN can also capture the local variability within each region much better than CloudSat, as demonstrated by Fig. 10 and Fig. 11 for the Antarctic Peninsula and Greenland, respectively.





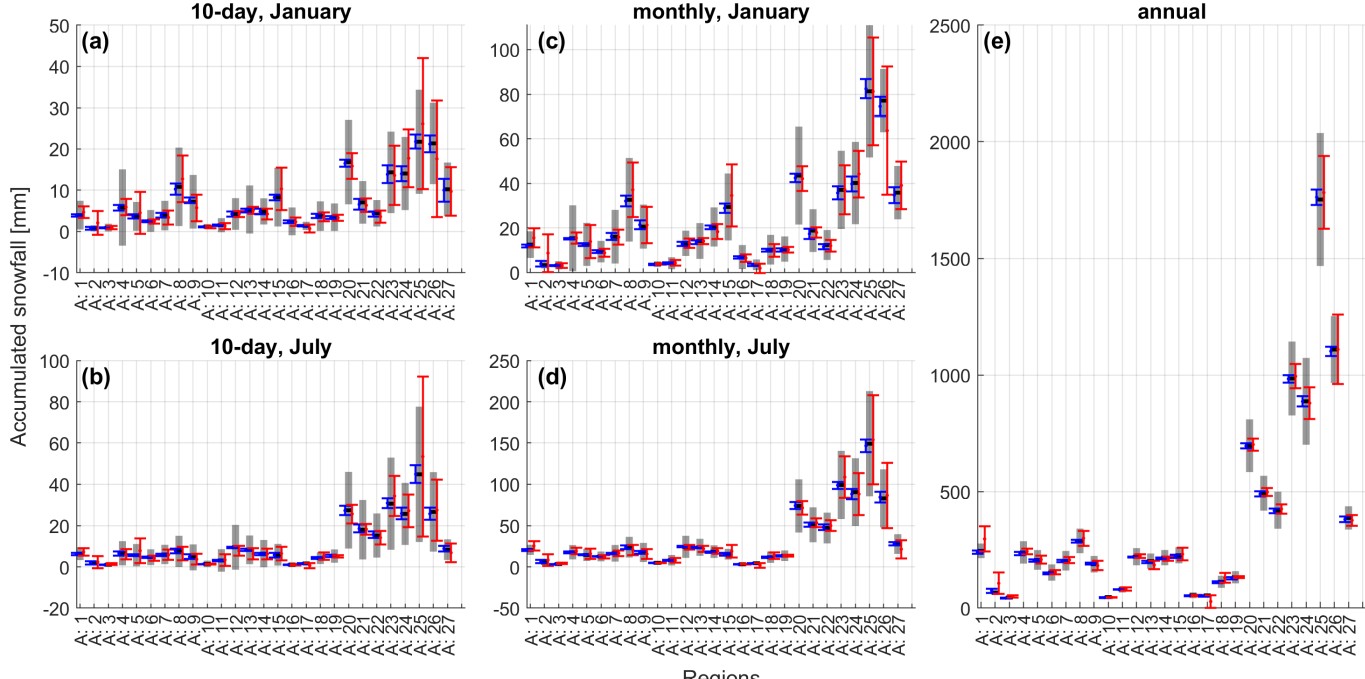

**Figure 8.** For each of the Antarctic regions (x-axis), the ERA5 mean snowfall accumulation (black line) and the climatological variability (grey shaded area) is shown. The mean accumulated snowfall ± RMSE sampled by WIVERN and CloudSat is shown with blue and red error bars, respectively. The result is shown for the 10-day, monthly and annual time scales in panels (a) and (b), (c) and (d), and (e) respectively. Results at the first two timescales are shown for January and July to highlight the different behavior between the two seasons.

## 4   Summary and conclusions

Spaceborne cloud radars are essential tools for observing snowfall globally (Stephens et al., 2018; Battaglia et al., 2020). The measurements are relevant for providing estimates of the snowfall accumulation, and thus for a wide range of applications: from

regional water cycle budgets to quantification of the mass balance changes of the ice sheet, the ice shelves and the glaciers. The reliability of such products is severely compromised by the intermittent and sparse sampling of snowfall carried out by the radar, with the number of samples collected in a given region in a given a time frame depending on the satellite orbit and on the radar scan geometry. For example, the WIVERN conically scanning radar (currently in Phase A of ESA's Earth Explorer programme) collects an order of magnitude more samples than a CloudSat or EarthCARE-like fixed near-nadir radar, which

also has gaps in coverage due to the narrow swath.

In this paper, the ERA5 hourly snowfall dataset has been used as a reference to simulate 20 years of snowfall accumulation as would have been sampled by a 94 GHz radar with WIVERN and CloudSat sampling geometry. Such accumulations are compared with the reference to assess the spatial and temporal scales at which these sensors become useful tools for estimating



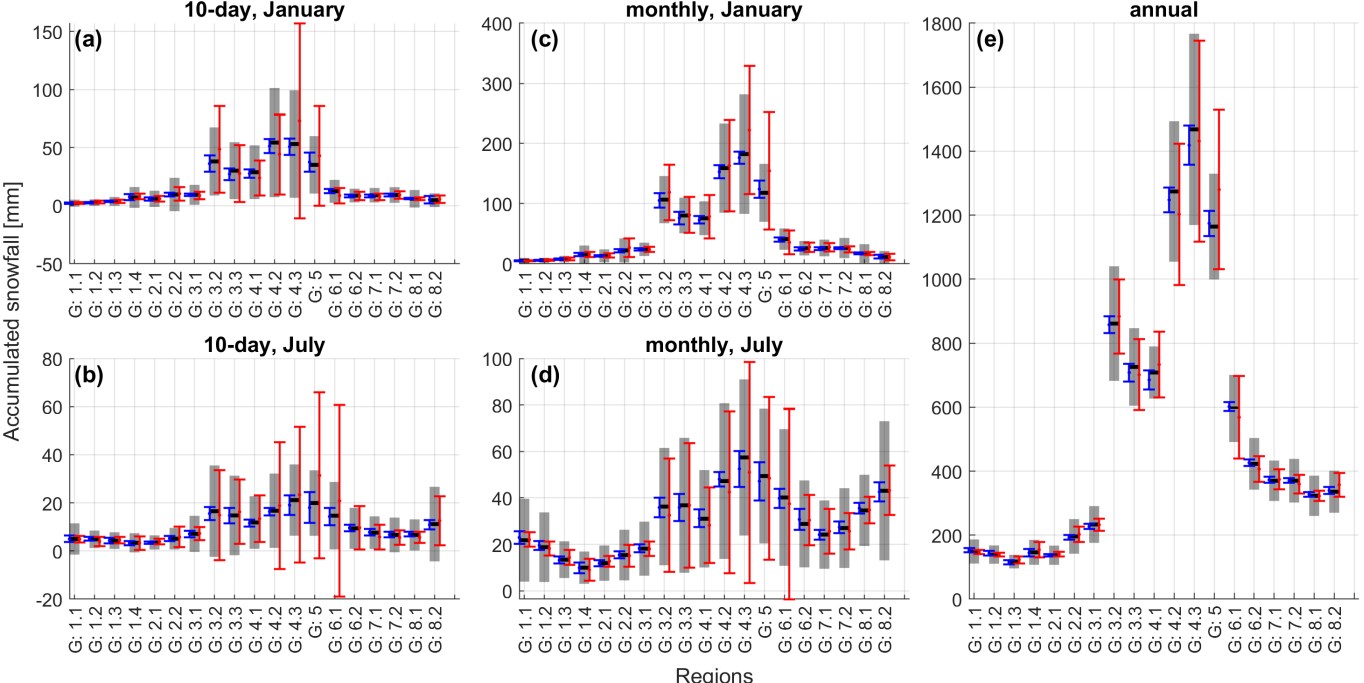

**Figure 9.** For each of the Greenland regions (x-axis), the ERA5 mean snowfall accumulation (black line) and the climatological variability (grey shaded area) is shown. The mean accumulated snowfall $\pm$ RMSE sampled by WIVERN and CloudSat is shown with blue and red error bars, respectively. The result is shown for the 10-day, monthly and annual time scales in panels (a) and (b), (c) and (d), and (e) respectively. Results at the first two timescales are shown for January and July to highlight the different behavior between the two seasons.

seasonal and/or regional accumulated snowfall. The error introduced by the two radars can be decomposed into the sampling error directly related to the intermittent sampling of the phenomenon, the error due to the uncertainty in the $Z_e - S$ relationship (assumed to be unbiased) and the error introduced by the minimum detectability threshold of the radar. Each contribution to the error has also been analyzed separately. To provide useful measurements, the error should be lower than the climatological variability, which is set to be the threshold of acceptable errors. The results show that:

– For WIVERN and CloudSat, the sampling error is the main cause of uncertainty. It decreases as the temporal and spatial
scale increase, with the error of WIVERN always beaing at least twice as small as the error of CloudSat.

– The error due to the $Z_e - S$ uncertainty is strongly mitigated as the spatial and temporal scale increase, as expected from the assumption of it being unbiased. For WIVERN, the large number of samples collected makes the error negligible starting from the $1.0° \times 1.0°$ spatial scale.



**Figure 10.** The top row shows the mean monthly and annual snowfall on the Antarctic Peninsula according to ERA5 on a lat-lon grid with box sizes of $0.25° \times 0.25°$. The middle and bottom rows show the corresponding NRMSE of WIVERN and CloudSat, respectively.

- – The radar sensitivity error is higher for WIVERN than for CloudSat ($-21$ vs. $-28$ dBZ) but the error is generally
 negligible, except in the regions where the snowfall rates are very low and constant in time for WIVERN (e.g. in the centre of Antarctica). As it is only relevant in correspondence of marginal snowy areas, its effect is globally insignificant.

- – Overall, WIVERN produces acceptable errors which are below the ERA5 climatological variability at all analyzed timescales at the $0.5° \times 0.5°$ spatial scales. Conversely, CloudSat needs to be averaged at annual zonal scales to produce reliable estimates.

 - – In the context of zonal snowfall, WIVERN and CloudSat are expected to correctly reproduce the mean value of the reference with errors lower than the climatological variability at the 10-day and annual scale, respectively.



**Figure 11.** The top row shows the mean monthly and annual snowfall on Greenland and Iceland according to ERA5 on a lat-lon grid with box sizes of $0.25° \times 0.25°$. The middle and bottom rows show the corresponding NRMSE of WIVERN and CloudSat, respectively. Note that, in July, the snowfall on the points above the ocean is very low and is characterized by very weak or very rare snowfall events, which cause the NRMSE being $\sim 1$ for WIVERN and CloudSat.

– In the context of assessing total accumulation in various regions of Antarctica and Greenland, WIVERN can provide estimates that fall within the climatological variability of the region at all the analyzed timescales. Instead, CloudSat offers less precise estimates, with RMSE exceeding the variability in some of the regions. Furthermore, when examining the local variability within these regions. CloudSat estimates are highly imprecise. In contrast, WIVERN's better sampling capabilities lead to significantly more precise estimations.

In conclusion, CloudSat is suitable for estimating snowfall accumulation over large areas and longer time scales (e.g. annual zonal), but its poor sampling capabilities limit the possibility to derive annual or monthly precipitation over domain smaller than zonal scales. The recently launched EarthCARE radar will face very similar sampling issues. On the other hand, a conically



scanning wide swath radar, such as the one proposed by the WIVERN team, could represent a unique observing system due to its improved sampling capabilities, contributing to the snowfall accumulation estimates over domains smaller than $0.5° \times 0.5°$ already at the 10-day timescale.

Future studies should quantify the effect of the WIVERN angle of incidence on the near-surface blind zone. Given the strong reduction of the normalised surface backscatter cross-section over the ocean at oblique angles of incidence (Battaglia et al.,
2017; Wolde et al., 2019), WIVERN is expected to reduce the blind layer over ocean surfaces compared to nadir-looking radars (Meneghini and Kozu, 1990). For sea ice and snow-covered surfaces, the reduction is more uncertain and the importance of the clutter may even increase. Finally, the WIVERN radar will have a low NEDT (Noise Equivalent Delta Temperature) radiometer mode and will provide (noisy) estimates of polarimetric variables such as differential reflectivity and differential phase shifts (Rizik et al., 2024). This could further improve the estimation of snowfall rates compared to nadir looking W-band radars and
needs to be investigated.

*Author contributions.* FES carried out the most analysis and wrote the text. AB and MM defined the project, wrote parts of the text and reviewed the work. SL reviewed the work.

*Competing interests.* At least one of the (co-)authors is a member of the editorial board of The Cryosphere.

*Acknowledgements.* This work was supported in part by the European Space Agency under the activity WInd VElocity Radar Nephoscope
(WIVERN) Phase A Science and Requirements Consolidation Study, ESA Contract Number 4000144120/24/NL/IB/ab. FES's work was conducted during and with the support of the Italian national inter-university PhD course in Sustainable Development and Climate change (link: www.phd-sdc.it). MM was supported by the German Research Foundation (DFG, Deutsche Forschungsgemeinschaft) Transregional Collaborative Research Center SFB/TRR 172 (Project ID 268020496). This research used the Mafalda cluster at Politecnico di Torino.

*Disclaimer.* Generated using Copernicus Climate Change Service information 2024. Neither the European Commission nor ECMWF is
responsible for any use that may be made of the Copernicus information or data it contains.



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
