# Peer review of "How to reduce sampling errors in spaceborne cloud radar-based snowfall estimates"

_EGUsphere, 2024_

## Author Comment (AC1)

**Reply to Referee #1**

February 1, 2025

We thank the reviewer for his insightful and useful comments. We will address them to improve the manuscript. Below, you will find the reviewers comments in red and our reply in black.

**1 Referee #1: Major Comments**

- **Zonal and Regional Analysis**:
  The paper highlights WIVERN's superior sampling capabilities, but the interpretation of regional results (e.g., Antarctic basins) could benefit from more detailed discussion. For example, if the temporal resolution WIVERN offers is crucial for mass balance studies in these regions.

  Typically, annual and greater scales are useful to understand the ice sheet response to multi year climate modes (e.g. El Niño), ice flow changes due to long term melting or thickening, the impact of ice sheet melting on the sea level rise and the interannual variability of the ice sheet mass balance.
  On the other hand, daily to seasonal time scales are useful to understand the seasonal variability, the grounding line migrations and short-term oceanic or atmospheric forcing.

  We will expand the discussion on regional results, including a short explanation on how the temporal resolution is useful for mass balance studies.

- **Clarity of Figures and Tables:**
  While figures support the findings, some lack detailed captions or sufficient detail to differentiate WIVERN and CloudSat results effectively. Explaining key trends (e.g., differences in RMSE across snowfall classes) in the text accompanying each figure would improve clarity.

  We will include more details in the explanation of the results and figures.

- **Future Prospects and Limitations:**
  While the article touches on potential future research directions, it does not fully address the limitations of the current analysis (e.g., assumptions about the unbiased nature of the reflectivity-snowfall relationship). Adding this discussion would provide balance.

  We agree with the comment and will better discuss the limitations of the current analysis, such as the assumption of the Z-S relationship being unbiased, the assumption of not having a greater path integrated attenuation due to the slant view geometry, and the fact of having a stronger ground clutter over land which might reduce the number of shallow events observed by WIVERN.

- **Improved Approach for Error Statistics**
  The methodology for computing error statistics (e.g., RMSE, Absolute Bias) appears to be based on instantaneous measurements, i.e., snowfall rates derived from ERA5 at specific times corresponding to satellite overpasses, with modifications to account for uncertainties in the Z-S relationship. If this interpretation is correct, the resulting error distribution will follow a Gaussian distribution with zero mean (assuming unbiased error) and a standard deviation equal to the uncertainty in the Z-S relationship. Consequently, the RMSE decreases as $1/\text{sqrt(N)}$ by definition due to increased sampling, but this approach does not directly assess how sampling impacts the derived climatology. To evaluate the sampling's effect on climatological estimates, a different approach should be considered:

  - Define the Observing Period: For example, consider a one-month observation period.

– Compute Monthly Accumulations: Use the ERA5 hourly dataset to calculate the "true" monthly snowfall accumulation (or mean snow rate) for each grid cell.

– Compare with Sparse Spaceborne Measurements: Derive monthly accumulations (or mean rates) from the spaceborne measurements using their more sparse sampling.

– Compute RMSE on Monthly Estimates: Calculate the RMSE by comparing the monthly ERA5 estimates with the monthly estimates derived from the spaceborne measurements.

This approach would more accurately quantify the impact of sampling on the derived climatology, as it evaluates errors at the monthly scale rather than relying solely on instantaneous measurements.

We agree with the referee, but we think we adopted the same approach he described as more accurate to evaluate the sampling's effect. Specifically, we followed this procedure:

– Define the observing period: e.g. one month observation period.

– Compute the monthly accumulations: use the ERA5 hourly dataset to calculate the "true" (or reference) monthly snowfall accumulation (or mean snow rate) for each grid cell.

– For each time instant of the ERA5 hourly dataset (i.e. for each hour), compute a mask that flag each point of the grid whether it is observed by CloudSat or not within that hour. The same has been done for WIVERN.

– Hourly masks are applied to the ERA5 hourly dataset to derive the hourly observations of CloudSat and WIVERN.

– Monthly accumulations derived from spaceborne measurements are obtained by summing multiple hourly spaceborne measurements.

– Calculate the RMSE on monthly estimates by comparing the monthly ERA5 estimates with the monthly estimates derived from the spaceborne measurements.

Based on our understanding of the approach suggested by the referee, we think the methodology he suggested is the same as we adopted, and we will explain it better in the manuscript. Please, let us know if this is not true, as we may have misunderstood the methodology you suggested.

- **Misleading Implications About Shallow Snowfall Detection**:
  The article primarily focuses on snowfall estimation over land, which is critical for ice sheet mass balance studies. However, the introduction gives a misleading impression that WIVERN will observe more shallow snowfall events than CloudSat. While this may hold true over open oceans due to reduced surface clutter at slanted incidence angles, it is not the case over land or sea ice. This distinction is crucial and should be clarified to avoid overestimating WIVERN's capabilities in these contexts. Addressing this limitation upfront would align reader expectations with the radar's realistic performance in various environments.

[Figure]

[Figure]

Figure 1: Height of the Signal To Clutter Ratio (SCR) equal to 5dB for different values of $Z_e$ of the hydrometeors at the surface. Whisker plot shows the median, the $\pm 15^{\text{th}}$ and the $\pm 15^{\text{th}}$ values.

Another study computed the height of the Signal To Clutter Ratio (SCR) = 5 dB for WIVERN, CloudSat and EarthCare, for different classes of hydrometeor reflectivity at the surface ($Z_{[0-1]km}^{hydro}$).

These are shown in figure 1. Boxplots represent the median, 15th and 85th percentile values. Over ocean (left panel), the impact of WIVERN clutter is weaker than the one of CloudSat and EarthCare, especially at larger $Z^{hydro}_{[0-1]km}$. Instead, over land (right panel), WIVERN clutter is slightly stronger, than the other two. We will make this limitation clearer to the readers, and we will address it also earlier in the manuscript.

- **Complexity and Clarity in Figure 4**:
  Figure 4 presents complex data, and the description lacks sufficient detail to make it accessible to readers. Key concepts, such as the definition of accumulation classes, need clarification. For instance:

  - Do the accumulation classes (e.g., snowfall between 36 and 108 mm per year) correspond to snowfall accumulation over the specified period (10 days, one month, or one year) in the ERA5 dataset?
  - How is the ERA5 variability computed for these classes? Is it the standard deviation of snowfall rates for grid cells corresponding to the specified range or maybe it's a mean of the normalised standard deviations?

  Although the concept behind the figure is straightforward, the lack of a detailed explanation makes it harder to follow. Additionally, referencing the central limit theorem (https://en.wikipedia.org/wiki/Central_limit_theorem) could greatly simplify the discussion. The explanation could say that the PDF being sampled is the ERA5 hourly snowfall product for each pixel separately, and the difference in sampling (WIVERN with n1 samples vs. CloudSat with n2, where n2<n1) leads to RMSE convergence as std(snow rate)/sqrt(n) when n is large. This statistical insight could make the sampling error analysis more intuitive. As the domain size or sampling time window grows the value of n grows too and RMSE decreases. The RMSE will be additionally inflated by the S-Z relationship uncertainty but this will affect both instruments in the same way as n gets larger.

  The accumulation classes correspond to the snowfall accumulation over the specified period, in the corresponding ERA5 dataset sampled by WIVERN or CloudSat, and averaged over the given spatial scale domain.
  The variability is computed as the standard deviation of the snowfall rates in all the grid cells across 20 years of data having snowfall rates belonging to a given class, divided by the mean of the snowfall rates of the same grid cells.
  We agree the figure may lack of clarity and we will improve the description of the figure. We will also reference to the central limit theorem when explaining the obtained RMSE results, as the reviewer suggests.

- **Blind zone effect**
  To provide a complete assessment, the paper should include an analysis of how ground clutter affects snowfall statistics for both WIVERN and CloudSat. Currently, this aspect is not addressed, which leaves a significant gap in understanding the limitations of these radar systems. While deriving these statistics directly from ERA5 data would be ideal, it would require extensive effort to analyse the vertically resolved precipitation product. A practical alternative would be to use statistics from the DAR-DAR (Radar-Lidar) A-train product. By deriving a 2D PDF of surface precipitation rate versus cloud top height, the authors could simulate the probability of an event being captured by both radars. This could be done by randomly sampling from the derived PDF. Events with weather system tops falling below the clutter height should have their precipitation set to zero, similar to the treatment in the radar sensitivity discussion. This approach would provide an insightful comparison of WIVERN and CloudSat performance, accounting for ground clutter effects. Obviously, it would not account for processes below the ground clutter height but it will provide a more comprehensive picture of radar limitations in snowfall detection.

  We will use data shown in figure 1 and discussed in the fifth major comment in synergy with DAR-DAR product, as suggested by the reviewer, to evaluate the ground clutter effects on snowfall statistics. We will then include the analysis and results in the next revised version of the manuscript.

**2   Referee #1: Minor Comments**

- Due to the orbit repeat cycle of 25 days, the analysis of CloudSat data blow this time resolution period has limited value.

  We will work to remove the 10 day resolution period and include the seasonal time-scale.

- L106: pencil beam term is used for the delta distribution, use "nadir observations" instead.

  We will substitute the two expressions.

- L190: units of snowfall rate should be mm/h

  We will correct it.

- Figure 7. Some of the colours in the colour bar are repeated or too similar to be distinguishable (e.g. two shades of grey: G7.2 and G3.3 or red G1.2 and G4.1, please use hatching or another way to make them less alike)

  We will improve the colorbar to make the plot clearer.

---

## Author Comment (AC2)

**Reply to Referee #2**

February 1, 2025

We thank the reviewer for his insightful and useful comments. We will address them to improve the manuscript. Below, you will find the reviewers comments in red and our reply in black.

**1   Referee #2: Minor Comments**

1. p3, line 63 - The authors refer to a snowfall rate in mm per hour. Is this liquid equivalent? If so, please be specific.

   Yes, it is water equivalent. We will specify it in the text.

2. p3, line 70 - Here you report an angle of 38 degrees, but above you report an incidence angle of 42 degrees. Please resolve this apparent discrepancy.

   38 degrees is the off-nadir angle, computed with respect to the nadir direction. Instead, 42 degrees is the incidence angle, computed from the zenith direction in the point where the boresight intersects the Earth's surface. Both are correct. We will specify there that 38 degrees is the off nadir angle.

3. pp4-5, lines 88-103 - Do you draw uncertainties from a Log-normal distribution because of the assumed power law relationship? Please add more detail as to how you include this noise. Do you run a Monte Carlo experiment? If so, how many samples do you use?

   We assumed S is log-normal distributed in order to have Z in dBZ normal distributed. Yes, it's like we ran a Monte Carlo experiment, where, for each point of the hourly LatLon grid, we sampled one value of S from the lognormal distribution. If we would have repeated the experiment many times, the samples of S would have been log-normal distributed.

4. pp4-5, lines 88-103 - Would you expect there to be bias in the Z-S relationship?

   Yes, typically, the Z-S relationship is biased, and literature tries to find unbiased relationships. However, Z-S cannot be unbiased in all applications, contexts and regimes. They need to be tuned for the specific appliations. The Z-S being unbiased is a strong assumption we made, but we assume that unbiased Z-S relationships can be found for specific regimes and locations via ground validation campaigns or complementary satellite observations.

5. pp4-5, lines 88-103 - The distributions in Hiley et al., 2011 were generated by sampling multiple crystal shapes and also allowing a limited range of PSD variability (via the temperature dependence of the Field et al. 2005 parameterization). Can you comment on whether there might be other sources of uncertainty (e.g., the fact that rimed aggregates and graupel were not considered in the scattering calculations)?

   Yes, as the Hiley Z-S relationships are retrieved at specific regimes (e.g. no riming and graupel is considered). Outside those regimes, (e.g. in presence of riming, presence of supercooled particles, presence of other particles shapes, ...) other sources of uncertainty might exist but WIVERN will also have a radiometric mode that could identify the presence of riming and/or supercooled layers.

6. pp10-11, lines 187-196 - Is there also an effect of the increased attenuation / path length due to 38 degree view angle (vs nadir view from CloudSat) on the WIVERN snowfall estimates?

Yes, there will be an effect due to the increased attenuation of WIVERN with respect to CloudSat due to the slant view angle which has not been included in our analysis but this contribution will only marginally reduce the SNR at very high snow rates. We will include this comment in the text.

7. pp. 14-18 - Discussion/Conclusions: A significant advance in WIVERN is the Doppler capability, providing both horizontal and vertical (line of sight) motions. I imagine that the combination of cloud structure (and snow mass retrievals) with dynamics could improve snow estimates even further (above the already substantial benefit of improved sampling). I suggest including some text along these lines.

Although WIVERN Doppler measurements will provide the horizontal dynamic structure of snow-producing storms we believe that these measurements will not improve substantially snow estimates. Instead, the radiometric mode might help in doing this by, for instance, providing information on the presence of supercooled droplets.

---

## Author Comment (AC3)

**Reply to Referee #3**

January 31, 2025

We thank the reviewer for his insightful and useful comments. We will address them to improve the manuscript. Below, you will find the reviewers comments in red and our reply in black.

**1   Referee #3: Minor Comments**

1. Line 6: Citation(s) for the statement that CloudSat is considered the reference?

   We will add citations to Palerme et al., 2014, and Stephens et al., 2018 in the introduction.

2. Line 7: inability to see shallow or retrieve heavy precipitation.

   We will modify it.

3. Line 13: Reword "This radar measurements" - suggest "The proposed radar measurements"

   We will reword as suggested.

4. Line 29: Reword to "snowfall not only removes moisture..."

   We will reword as suggested.

5. Line 76: Please give some background to ERA5 snowfall. Validation, assimilated data, etc.

   We chose ERA5 snowfall dataset as a reference because it is considered to be one of the most accurate and precise reanalysis product. What is important for our study is that it provides realistic spatio-temporal variability of the snow fields. We can add some references to similar studies that uses the same dataset.

6. Figure 4: These plots are a bit dense and need more explanation or possibly simplification

   We will better explain the plots and will remove the histogram on the ERA5 occurrences to simplify the plots.

7. Line 215: Why not shown?

   We didn't include in the plots the zonal mean of the snowfall captured by CloudSat and Wivern to simplify the plots. We can include those in the revised version.

8. Line 220: Why not shown? Maybe not worth mentioning the 10-day time scale given CloudSat's sampling

   We didn't include in the zonal plot referred to the 10-day timescale because we thought it was not important since the repeat cycle of the CloudSat orbit is largerer than 10 days. We will substitute the 10-day timescale with the seasonal timescale in the revised version and will include the related plots.

9. Line 250: Please add more explanation/narrative here regarding the local variability plots

   We will change the colorbar limits of the plots to improve readability and will add some explanation to it. In the Antarctic Peninsula and Western Greenland there are some spots where snowfall is significantly larger than in the whole area. WIVERN can capture this at monthly and annual scales. Instead, as the figures show, CloudSat sampling do not provide sensible information on those hotspots.

10. Figures 8 and 9 would benefit from a legend

    We will add the legend,

---

## Author Response (AR1)

**Reply to Referees**

April 21, 2025

Please, find below our replies to the referees comments.

**1 Referee #1**

**1.1 Major comments**

**• Zonal and Regional Analysis:**

The paper highlights WIVERN's superior sampling capabilities, but the interpretation of regional results (e.g., Antarctic basins) could benefit from more detailed discussion. For example, if the temporal resolution WIVERN offers is crucial for mass balance studies in these regions.

Typically, estimates at annual and greater scales are useful to understand the ice sheet response to multi-year climate modes (e.g., El Niño), ice flow changes due to long term melting or thickening, the impact of ice sheet melting on the sea level rise and the inter-annual variability of the ice sheet mass balance. On the other hand, estimates at daily to seasonal time scales are useful to understand the seasonal variability, the grounding line migrations and short-term oceanic or atmospheric forcing.

We added the paragraph above in lines 253-257.

**• Clarity of Figures and Tables:**

While figures support the findings, some lack detailed captions or sufficient detail to differentiate WIVERN and CloudSat results effectively. Explaining key trends (e.g., differences in RMSE across snowfall classes) in the text accompanying each figure would improve clarity.

We tried to improve clarity of the figures (e.g. we added a better description for figure 4 at lines 175-187).

**• Future Prospects and Limitations:**

While the article touches on potential future research directions, it does not fully address the limitations of the current analysis (e.g., assumptions about the unbiased nature of the reflectivity-snowfall relationship). Adding this discussion would provide balance.

We added this discussion in the introduction (lines 75-88), and in the specific sections the limitations of this analysis: Z-S considered to be unbiased, no surface blind zone effect included, path integrated attenuation considered to be negligible.

**• Improved Approach for Error Statistics**

The methodology for computing error statistics (e.g., RMSE, Absolute Bias) appears to be based on instantaneous measurements, i.e., snowfall rates derived from ERA5 at specific times corresponding to satellite overpasses, with modifications to account for uncertainties in the Z-S relationship. If this interpretation is correct, the resulting error distribution will follow a Gaussian distribution with zero mean (assuming unbiased error) and a standard deviation equal to the uncertainty in the Z-S relationship. Consequently, the RMSE decreases as 1/sqrt(N) by definition due to increased sampling, but this approach does not directly assess how sampling impacts the derived climatology. To evaluate the sampling's effect on climatological estimates, a different approach should be considered:

- Define the Observing Period: For example, consider a one-month observation period.

- Compute Monthly Accumulations: Use the ERA5 hourly dataset to calculate the "true" monthly snowfall accumulation (or mean snow rate) for each grid cell.
- Compare with Sparse Spaceborne Measurements: Derive monthly accumulations (or mean rates) from the spaceborne measurements using their more sparse sampling.
- Compute RMSE on Monthly Estimates: Calculate the RMSE by comparing the monthly ERA5 estimates with the monthly estimates derived from the spaceborne measurements.

This approach would more accurately quantify the impact of sampling on the derived climatology, as it evaluates errors at the monthly scale rather than relying solely on instantaneous measurements.

We agree with the referee, but we think we adopted the same approach he described as more accurate to evaluate the sampling's effect. Specifically, we followed this procedure:

- Define the observing period: e.g. one month observation period.
- Compute the monthly accumulations: use the ERA5 hourly dataset to calculate the "true" (or reference) monthly snowfall accumulation (or mean snow rate) for each grid cell.
- For each time instant of the ERA5 hourly dataset (i.e. for each hour), compute a mask that flag each point of the grid whether it is observed by CloudSat or not within that hour. The same has been done for WIVERN.
- Hourly masks are applied to the ERA5 hourly dataset to derive the hourly observations of CloudSat and WIVERN.
- Monthly accumulations derived from spaceborne measurements are obtained by summing multiple hourly spaceborne measurements.
- Calculate the RMSE on monthly estimates by comparing the monthly ERA5 estimates with the monthly estimates derived from the spaceborne measurements.

Based on our understanding of the approach suggested by the referee, we think the methodology he suggested is the same as we adopted. Please, let us know if this is not true, as we may have misunderstood the methodology you suggested.

**• Misleading Implications About Shallow Snowfall Detection:**

The article primarily focuses on snowfall estimation over land, which is critical for ice sheet mass balance studies. However, the introduction gives a misleading impression that WIVERN will observe more shallow snowfall events than CloudSat. While this may hold true over open oceans due to reduced surface clutter at slanted incidence angles, it is not the case over land or sea ice. This distinction is crucial and should be clarified to avoid overestimating WIVERN's capabilities in these contexts. Addressing this limitation upfront would align reader expectations with the radar's realistic performance in various environments.

Figure 1: Height of the Signal To Clutter Ratio (SCR) equal to 5dB for different values of  $Z_e$  of the hydrometeors at the surface. Whisker plot shows the median, the  $\pm 15^{\rm th}$  and the  $\pm 15^{\rm th}$  values. Left and right panels are referred to ocean and land surfaces, respectively, (Coppola et al., 2025)

Coppola et al. (2025) computed the height of the Signal To Clutter Ratio (SCR) = 5 dB for WIVERN, CloudSat and EarthCare, for different classes of hydrometeor reflectivity at the surface  $(Z_{[0-1]km}^{hydro})$ . These are shown in figure 1. Boxplots represent the median, 15th and 85th percentile values. Over ocean (left panel), the impact of WIVERN clutter is weaker than the one of CloudSat and EarthCare, especially at larger  $Z_{[0-1]km}^{hydro}$ . Instead, over land (right panel), WIVERN clutter is slightly stronger, than the other two. We made this limitation clearer by discussing it in the introduction and in a dedicated section (section 4).

**• Complexity and Clarity in Figure 4:**

Figure 4 presents complex data, and the description lacks sufficient detail to make it accessible to readers. Key concepts, such as the definition of accumulation classes, need clarification. For instance:

- Do the accumulation classes (e.g., snowfall between 36 and 108 mm per year) correspond to snowfall accumulation over the specified period (10 days, one month, or one year) in the ERA5 dataset?
- How is the ERA5 variability computed for these classes? Is it the standard deviation of snowfall rates for grid cells corresponding to the specified range or maybe it's a mean of the normalised standard deviations?

Although the concept behind the figure is straightforward, the lack of a detailed explanation makes it harder to follow. Additionally, referencing the central limit theorem (https://en.wikipedia.org/wiki/Central\_limit\_theorem) could greatly simplify the discussion. The explanation could say that the PDF being sampled is the ERA5 hourly snowfall product for each pixel separately, and the difference in sampling (WIVERN with n1 samples vs. CloudSat with n2, where n2<n1) leads to RMSE convergence as std(snow rate)/sqrt(n) when n is large. This statistical insight could make the sampling error analysis more intuitive. As the domain size or sampling time window grows the value of n grows too and RMSE decreases. The RMSE will be additionally inflated by the S-Z relationship uncertainty but this will affect both instruments in the same way as n gets larger.

The accumulation classes correspond to the snowfall accumulation over the specified period, in the corresponding ERA5 dataset sampled by WIVERN or CloudSat, and averaged over the given spatial scale domain.

The variability is computed as the standard deviation of the snowfall rates in all the grid cells across 20 years of data having snowfall rates belonging to a given class, divided by the mean of the snowfall rates of the same grid cells.

We added these clarifications and a reference to the central limit theorem.

**• Blind zone effect**

To provide a complete assessment, the paper should include an analysis of how ground clutter affects snowfall statistics for both WIVERN and CloudSat. Currently, this aspect is not addressed, which leaves a significant gap in understanding the limitations of these radar systems. While deriving these statistics directly from ERA5 data would be ideal, it would require extensive effort to analyse the vertically resolved precipitation product. A practical alternative would be to use statistics from the DAR-DAR (Radar-Lidar) A-train product. By deriving a 2D PDF of surface precipitation rate versus cloud top height, the authors could simulate the probability of an event being captured by both radars. This could be done by randomly sampling from the derived PDF. Events with weather system tops falling below the clutter height should have their precipitation set to zero, similar to the treatment in the radar sensitivity discussion. This approach would provide an insightful comparison of WIVERN and CloudSat performance, accounting for ground clutter effects. Obviously, it would not account for processes below the ground clutter height but it will provide a more comprehensive picture of radar limitations in snowfall detection.

We used the estimated signal to clutter ratio = 5dB estimated for CloudSat and WIVERN in Coppola et al. (2025) (shown in figure 1 of this document) in synergy with ERA5 vertical snow water content profiles, to evaluate the ground clutter effects on snowfall statistics. We included the analysis, its limitations, and the results and in section 4 of the new version of the manuscript.

We also tried with DAR-DAR but the profiles looked unrealistic in the last 1 km near the surface, probably because they suffer from the same clutter as CloudSat.

**1.2 Minor Comments**

• Due to the orbit repeat cycle of 25 days, the analysis of CloudSat data blow this time resolution period has limited value.

For this reason, we removed the 10 days timescale and added the seasonal timescale.

- L106: pencil beam term is used for the delta distribution, use "nadir observations" instead. We changed the terms.
- L190: units of snowfall rate should be mm/h We corrected it.
- Figure 7. Some of the colours in the colour bar are repeated or too similar to be distinguishable (e.g. two shades of grey: G7.2 and G3.3 or red G1.2 and G4.1, please use hatching or another way to make them less alike)

We improved the colorbars and added the labels in the figure on each sub-region.

**2 Referee #2**

**2.1 Minor comments**

• p3, line 63 - The authors refer to a snowfall rate in mm per hour. Is this liquid equivalent? If so, please be specific.

Yes, it is water equivalent. We added this clarification in the text.

- p3, line 70 Here you report an angle of 38 degrees, but above you report an incidence angle of 42 degrees. Please resolve this apparent discrepancy.
  - 38 degrees is the off-nadir angle, computed with respect to the nadir direction in the spacecraft location. Instead, 42 degrees is the incidence angle, computed with respect to the surface in the point where the boresight intersects the Earth's surface. Both are correct. We will specify there that 38 degrees is the off nadir angle.
- pp4-5, lines 88-103 Do you draw uncertainties from a Log-normal distribution because of the assumed power law relationship? Please add more detail as to how you include this noise. Do you run a Monte Carlo experiment? If so, how many samples do you use?

We assumed S is log-normal distributed in order to have Z in dBZ normal distributed ( we added this clarification in the text). Yes, it's like we ran a Monte Carlo experiment, where, for each point of the hourly LatLon grid, we sample one value of S from the lognormal distribution.

• pp4-5, lines 88-103 - Would you expect there to be bias in the Z-S relationship?

Yes, typically, the Z-S relationship is biased, and literature tries to find unbiased relationships. However, Z-S cannot be unbiased in all applications, contexts and regimes. They need to be tuned for the specific appliations. The Z-S being unbiased is a strong assumption we made, but we assume that unbiased Z-S relationships can be found for specific regimes and locations via ground validation campaigns or complementary satellite observations. We added this in the introduction (lines 84 and 85) and in lines 119, 120.

• pp4-5, lines 88-103 - The distributions in Hiley et al., 2011 were generated by sampling multiple crystal shapes and also allowing a limited range of PSD variability (via the temperature dependence of the Field et al. 2005 parameterization). Can you comment on whether there might be other sources of uncertainty (e.g., the fact that rimed aggregates and graupel were not considered in the scattering calculations)?

Yes, as the Hiley Z-S relationships are retrieved at specific regimes (e.g. no riming and graupel is considered). Outside those regimes, (e.g. in presence of riming, presence of supercooled particles, presence of other particles shapes, ...) other sources of uncertainty might exist but WIVERN will also have a radiometric mode that could identify the presence of riming and/or supercooled layers.

We added this in lines 119-121.

• pp10-11, lines 187-196 - Is there also an effect of the increased attenuation / path length due to 38 degree view angle (vs nadir view from CloudSat) on the WIVERN snowfall estimates?

Yes, there will be an effect due to the increased attenuation of WIVERN with respect to CloudSat due to the slant view angle which has not been included in our analysis but this contribution will only marginally reduce the SNR at very high snow rates.

We added this comment in the introduction (lines 86-87).

• pp. 14-18 - Discussion/Conclusions: A significant advance in WIVERN is the Doppler capability, providing both horizontal and vertical (line of sight) motions. I imagine that the combination of cloud structure (and snow mass retrievals) with dynamics could improve snow estimates even further (above the already substantial benefit of improved sampling). I suggest including some text along these lines.

Although WIVERN Doppler measurements will provide the horizontal dynamic structure of snow-producing storms we believe that these measurements will not improve substantially snow estimates. Instead, the radiometric mode might help in doing this by, for instance, providing information on the presence of supercooled droplets. This comment has been added in the conclusions.

**3 Referee #3**

**3.1 Minor comments**

• Line 6: Citation(s) for the statement that CloudSat is considered the reference?

We will add citations to Palerme et al., 2014, and Stephens et al., 2018 in the introduction.

Line 7: inability to see shallow or retrieve heavy precipitation.
 We modified it.

• Line 13: Reword "This radar measurements" - suggest "The proposed radar measurements" We reworded as suggested.

• Line 29: Reword to "snowfall not only removes moisture..."

We reworded as suggested.

• Line 76: Please give some background to ERA5 snowfall. Validation, assimilated data, etc.

We chose ERA5 snowfall dataset as a reference because it is considered to be one of the most accurate and precise reanalysis product. What is important for our study is that it provides realistic spatio-temporal variability of the snow fields. We included a reference to Kouki et. al, 2023.

• Figure 4: These plots are a bit dense and need more explanation or possibly simplification

We improved the explanation of the plots and removed the ERA5 histogram from the plots, which was not necessary, to simplify the plots.

• Line 215: Why not shown?

Because there is no departure from the ERA5 mean, However, we added them (zonal means of WIVERN and CloudSat) to the figure of the new version of the manuscript.

• Line 220: Why not shown? Maybe not worth mentioning the 10-day time scale given CloudSat's sampling

We didn't include in the zonal plot referred to the 10-day timescale because we thought it was not important since the repeat cycle of the CloudSat orbit is larger than 10 days. We substituted the 10-day timescale with the seasonal timescale.

• Line 250: Please add more explanation/narrative here regarding the local variability plots

In the Antarctic Peninsula and Western Greenland there are some spots where snowfall is significantly larger than in the whole area. WIVERN can capture this at monthly and annual scales. Instead, as the figures show, CloudSat sampling do not provide sensible information on those hotspots. We added this comment in the manuscript (lines 284-286).

• Figures 8 and 9 would benefit from a legend We added the legend.